# Serological Response to SARS-CoV-2 Messenger RNA Vaccine: Real-World Evidence from Italian Adult Population

**DOI:** 10.3390/vaccines9121494

**Published:** 2021-12-17

**Authors:** Rosa Papadopoli, Caterina De Sarro, Caterina Palleria, Luca Gallelli, Claudia Pileggi, Giovambattista De Sarro

**Affiliations:** Department of Health Sciences, Campus “Salvatore Venuta”, University of Catanzaro “Magna Græcia”, 88100 Catanzaro, Italy; rosapapadopoli@unicz.it (R.P.); caterina.desarro@studenti.unicz.it (C.D.S.); palleria@unicz.it (C.P.); gallelli@unicz.it (L.G.); desarro@unicz.it (G.D.S.)

**Keywords:** anti-spike antibodies, messenger RNA vaccine, neutralizing antibody response, SARS-CoV-2, seroconversion

## Abstract

Background: This study aims to investigate the extent of the BNT162b2 mRNA vaccine-induced antibodies against SARS-CoV-2 in a large cohort of Italian subjects belonging to the early vaccinated cohort in Italy. Methods: A prospective study was conducted between December 2020 and May 2021. Three blood samples were collected for each participant: one at the time of the first vaccine dose (T0), one at the time of the second vaccine dose, (T1) and the third 30 days after this last dose (T2). Results: We enrolled 2591 fully vaccinated subjects; 16.5% were frail subjects, and 9.8% were over 80 years old. Overall, 98.1% of subjects were seropositive when tested at T2, and 76.3% developed an anti-S IgG titer ≥4160 AU/mL, which is adequate to develop viral neutralizing antibodies. Seronegative subjects at T1 were more likely to remain seronegative at T2 or to develop a low–intermediate anti-S IgG titer (51–4159 AU/mL). Conclusions: In summary, vaccination leads to detectable anti-S IgG titer in nearly all vaccine recipients. Stratification of the seroconversion level could be useful to promptly identify high-risk groups who may not develop a viral neutralizing response, even in the presence of seroconversion, and therefore may remain at higher risk of infection, despite vaccination.

## 1. Introduction

A new acute respiratory syndrome, coronavirus disease 2019 (COVID-19), emerged from the region of Wuhan, China in December 2019. The causative pathogen was subsequently identified as the severe acute respiratory syndrome related coronavirus 2 (SARS-CoV-2) [1], with severe morbidity and mortality rates [2,3].

This virus, now recognized as the etiologic agent of COVID-19 disease, is the seventh known coronavirus to infect humans [4]. Since the recognition of COVID-19, there has been an exponential rise in the number of cases worldwide. The disease SARS-CoV-2 has spread swiftly to over 200 countries, infecting almost 200 million people and resulting in more than 4,000,000 deaths worldwide [5].

Reasons for the rapid spread include high transmissibility of the virus [6], especially among asymptomatic or minimally symptomatic carriers [7,8,9], and the apparent absence of any cross-protective immunity from related viral infections.

To date, several treatments have been indicated for the management of COVID-19 infection, with the principal aim of reducing the burden of SARS-CoV-2 on patients, alleviating severe symptoms and slowing the progress of COVID-19 [10], even if a definitive treatment has not yet been indicated [11].

Preventive vaccination is the safest and most cost-effective way to prevent COVID-19 illness and death, and the best option to combat anticipated future variants. Several vaccines have been developed including molecular, particular and cell-based vaccines since the beginning of the pandemic. The BNT162b2 mRNA vaccine developed by BioNTech and Pfizer produced the first SARS-CoV-2 vaccine that was approved in Italy, which has been shown to induce anti-spike (S), protein-specific antibodies (anti-S).

Definition of a vaccine’s efficacy depends on several factors, such as the characteristics of the pathogen, the consequences of infection and the dynamics of transmission. In the context of a novel pathogen such as SARS-CoV-2, evaluating vaccine efficacy is particularly complex, due to the evolving understanding of the pathogen. Efficacy outcomes for SARS-CoV-2 vaccines include clinical end-points, e.g., protection against severe disease and mortality, and a reduction in hospital admissions, as well as surrogate end-points such as the immunological correlates of protection. Antibodies developed after BNT162b2 mRNA vaccination seem to correlate with a certain grade of protection and prevention of viral activity [12]. Therefore, vaccine efficacy can be measured by the proportion of vaccinees who develop a particular immune response, thus avoiding the need to measure clinical outcomes. The antibody responses against SARS-CoV-2 are a marker of exposure, but whether or not these antibodies are adequate to provide protection against disease or infection following natural infection or vaccination is unclear [13,14]. Recently, Ebinger, J.E. et al. assessed in a large cohort of BNT162b2 mRNA vaccinees the neutralization potential of elicited antibodies, identifying the anti-S IgG titer ≥4160 AU/mL as a threshold which correlates with a 95% probability of developing neutralizing antibodies [15].

The primary aim of the study was to investigate, through serological analysis, the extent to which the BNT162b2 vaccine induced antibodies against SARS-CoV-2 in a large cohort of Italian subjects belonging to the cohort of early vaccine recipients in Italy.

## 2. Materials and Methods

### 2.1. Study Design and Population

A prospective study was conducted between December 2020 and May 2021, enrolling all of the subjects attending the COVID-19 vaccination center of the Teaching Hospital of the University “Magna Graecia” of Catanzaro (southern Italy). Study subjects included those identified as priority groups for SARS-CoV-2 vaccination by the Italian Ministry of Health (healthcare workers, frail patients and subjects aged 80 years or older) [16] and teaching and administrative University staff.

After the enrolment, subjects signed the informed consent documents and underwent a venous blood sample for serological analysis. Three blood samples were collected for each participant: one at the time of the first vaccine dose (baseline—T0), one after 21 days, at the time of the second vaccine dose (T1) and the third 30 days after this last dose (T2).

### 2.2. Laboratory Methods

Blood samples were transported to the laboratory and, after centrifugation, were stored at 4 °C until analysis. The detection of antibodies was performed within 48 h from the collection of the samples, using both SARS-CoV-2 IgM and IgG assay (Abbot Diagnostics-USA, Chicago, IL, USA) and Architect I1000 instrument (Abbot Diagnostics-USA). Sera collected at T0 were processed using IgM and IgG qualitative tests that detect antibodies targeting the viral nucleocapsid protein (N), and results equal or above 1.00 and 1.4 indicated the presence of IgM and of IgG antibodies, respectively. Quantitative post-vaccination anti-spike antibody responses, at times T1 and T2, were measured using the Abbott SARS-CoV-2 IgG II Quant assay with a cut-off of 50 Arbitrary Units (AU)/mL. Subjects who had previously contracted the SARS-CoV-2 infection and those with a baseline IgM and/or IgG titer equal or above the cut-off were classified as seropositive, and thus were excluded from the study.

According to Ebinger et coll. [15], a value greater than, or equal to 4160 AU/mL was used to evaluate the capacity of the recipients to achieve an anti-S IgG titer that would be adequate to develop viral neutralizing antibodies. Participants were then classified, according to their anti-S IgG titer at T2, into three groups: seronegative (≤50 AU/mL), low–intermediate responders (51–4159 AU/mL) and high responders (≥4160 AU/mL).

### 2.3. Statistical Analysis

Data were stored and analyzed using an appropriate database. Statistical analysis was performed using the STATA software program, version 16 (Stata Corporation, College Station, TX, USA).

Data were summarized using frequencies and percentages for categorical data, and means and standard deviations (SD) were used for continuous data. Demographic characteristics were compared using chi-square or Fisher’s exact tests for categorical variables, and the ANOVA test was used for continuous variables.

The outcome of interest was the anti-S IgG titer, measured 30 days after the full course of BNT162b2 vaccination, which was classified into three levels: ≤50 AU/mL anti-S titer (seronegative), 51–4159 AU/mL anti-S titer (low–intermediate responders) and ≥4160 AU/mL anti-S titer (high responders).

Finally, a multinomial logistic regression model was developed to determine the potential role of the independent variables (gender, age, risk category and anti-S IgG titer at T1) on the outcome variable. The following three risk categories were identified: (1) healthy subjects, including HCWs, teaching and administrative staff who did not suffer any illness; (2) frail subjects, defined in accordance with Hanlon, P. et al. as subjects affected by a condition identifiable in middle-aged as well as older adults, associated with a range of long-term conditions and with mortality independently of the extent of multimorbidity as well as age, sex, sociodemographic, and lifestyle factors; (3) subjects aged over 80 years [17].

Relative risk ratios (RRRs) and a 95% confidence interval (CI) were calculated, all reported *p* values were two-tailed, and a value <0.05 was considered statistically significant.

The study protocol was ratified by the Regional Ethical Committee (ID N.183).

## 3. Results

The eligible population was composed of 3000 subjects. Of these, 2631 (88%) provided the consent to undergo at least one blood sample for antibody testing. Forty subjects were found to be seropositive at baseline and were excluded, so the final sample included 2591 individuals. All participants were of white ethnic groups. The main characteristics of the study population are reported in Table 1. 

The age of participants ranged between 18 and 99 years (with a median of 50 years) and the majority were females (53.4%). Among the 1910 healthy subjects, 75.3% were permanent healthcare workers, 9.6% were undergraduate healthcare workers in training and 24.7% were teaching and administrative personnel. Of the included subjects, 16.5% were classified as frail patients due to the presence of at least two comorbidities. Blood samples were provided by 1725 (66.6%) subjects at T0, 1641 (63.3%) at T1, and by 2591 of subjects at T2, whereas 1215 (46.9%) subjects provided samples at all three time points. Overall, 2541/2591 (98.1%) subjects were seropositive when tested 30 days after the conclusion of the full course of vaccination; of these, 1939/2541 (76.3%) developed an anti-S IgG titer ≥4160 AU/mL. When the subgroup of healthy subjects was considered, only two participants were seronegative after the second dose with a percentage of seroconversion (anti-S IgG > 50 AU/mL) of 99.8%. Percentages of seroconversion (anti-S IgG > 50 AU/mL) stratified by gender and age categories are reported in Figure 1.

Results of the univariate analysis (Table 1) highlighted that seroconversion with a neutralizing antibodies titer (anti-S IgG titer ≥ 4160 AU/mL) was significantly more likely in females (χ^2^ = 52.1, 2 df, *p* < 0.001) and in younger subjects (F = 211, 08, 2 df, *p* < 0.001). Among the 1641 subjects who were tested at T1, 162 (9.9%) were seronegative (anti-S IgG ≤ 50 AU/mL) and, of these, 25 (15.4%) remained seronegative 30 days after the full vaccination course (T2), and 110 (67.9%) developed a low–intermediate anti-S IgG titer (51–4159 AU/mL). Moreover, a significantly higher anti-S IgG titer was found in subjects whose results were seropositive at T1 (Fisher exact *p* < 0.001), with 1236 (83.6%) subjects who achieved a ≥ 4160 AU/mL anti-S IgG titer. A seronegative titer (anti-S IgG ≤ 50 AU/mL) was significantly more frequent in subjects classified as frail (Fisher exact *p* < 0.001). However, the multinomial logistic regression model (Table 2) showed that only having a seronegative anti-S IgG titer (≤50 AU/mL) at T1 was significantly associated with no seroconversion (anti-S IgG ≤ 50 AU/mL) (RRR = 0.94, 95% CI-0.90–0.97) and with a low–intermediate response (51–4159 AU/mL) (RRR = 0.99, 95% CI-0.98–0.98), compared to the adequate response required to develop viral neutralizing antibodies (anti-S IgG ≥ 4160 AU/mL). 

## 4. Discussion

The results of this prospective study provide real-world data demonstrating that the BNT162b2 mRNA vaccine has induced seroconversion in the majority of a large cohort of vaccine recipients who may not participate in clinical trials.

Our study, characterized by a high participation rate, with 88% of subjects agreeing to undergo serological testing after being vaccinated, adds a relevant insight into the factors implicated in the seroconversion, indicating an important role of seroconversion after 21 days from the first vaccine dose. We found that the only predictor of the seroconversion after the full course of vaccination was the anti-S IgG titer measured at T1. Seronegative (anti-S IgG ≤ 50 AU/mL) subjects at T1 were more likely to remain seronegative at T2 or to develop a low–intermediate anti-S IgG titer (51–4159 AU/mL). Indeed, to analyze more thoroughly the effect of the BNT162b2 mRNA vaccination on seroconversion, we chose to separate subjects who showed an anti-S IgG titer ˃50AU/mL after the full vaccination course into two groups (low–intermediate responders: 51–4159 AU/mL and high responders: ≥4160 AU/mL), according to the recently suggested conservative threshold [15]. Almost a quarter of the total sample (23.2%) developed a low–intermediate response after the second vaccination dose and, as shown in the subgroup of subjects who provided blood sample at T1, this result was significantly more frequent among seronegative subjects after the first vaccine dose (67.9%), compared to those who were seropositive (16.3%). The role of the seroconversion level at T1 on the anti-S IgG titer after the second vaccine dose has also been reported in another recent study [18]. 

Our results may be useful to promptly identify a high-risk group who may not develop a viral neutralizing response, even in presence of seroconversion, and therefore may remain at a higher risk of infection despite vaccination, and may need additional vaccine doses, catch-up strategies and personalized advice on social contact. On the other hand, it is already known [19] that if seroconversion alone is not an exhaustive measure to define the level of immune protection, serial sampling at the correct timing correlates well with effective protective immunity, and our findings indicate their potential usefulness in guiding vaccine management strategies.

Several studies have investigated real-world antibody responses to the SARS-CoV-2 BNT162b2 vaccine in different categories, using different methods of ascertainment and reporting on seroconversion. Therefore, comparisons between the serological response to the BNT162b2 vaccine in our study and observations made from previous studies should be made cautiously. Our estimates of seroconversion among healthy subjects (99.8%) are consistent with the results of previous Italian studies which highlighted a seroconversion rate from 99.5% to 99.8% [20,21]. These results are also in line with those of Fraley, E. et al. [18], which showed a 99.2% seropositivity after vaccination in USA HCWs. Moreover, in a study from Israel, Abu Jabal, K. et al. reported early results from HCWs who, 21 days after the first vaccine dose, achieved 92% IgG seropositivity. This result is in line with our 90.1% [22]. A higher percentage of seroconversion after the first BNT162b2 mRNA vaccine dose has been shown by Eyre et coll. [14] with 99% of vaccine recipients developing a seropositive anti-S IgG titer. It should be noted that the median age of the study participants was 41 years, compared to the median of 50 years of our sample. Therefore, it is hypothesized that a younger age was the determinant of this more favorable outcome. Our findings are also consistent with those derived by a large study involving the UK general population [23] that showed a seropositivity of 90.8% 21 days after the first vaccine dose, and of 95.5% 30 days after the second. 

With regard to the frailty of subjects, previous studies on patients with chronic conditions [24,25,26,27] showed that they are able to develop an immunological response, although this response is lower than that reached in healthy subjects. Our results are consistent with the lower vaccine effectiveness that was reported in frail patients, with nearly half of subjects included in this subgroup remaining seronegative or developing a low–intermediate anti-S IgG titer after the full course of vaccination. Even if a statistically significant difference across risk categories was not reached in the multinomial regression analysis, the wide 95% CI (0.70 to 25.1) related to the RRR remained seronegative after the full course of vaccination, indicating that our study was not equipped to definitively assess effectiveness by subgroup. Therefore, it is necessary to recognize this concern within the limitations of the study. Furthermore, given the social setting of the SARS-CoV-2 vaccination campaign, convenience sampling inevitably led to missing data for the study end-points. Larger cohorts are needed for evaluating vaccine responses in subpopulations to overcome the inherent biases of low-powered studies.

Concerning the role of demographic characteristics in the antibody responses to SARS-CoV-2 vaccines, our results showed at the univariate statistical analysis a significantly higher titer in younger subjects and in women (although these results were not confirmed in the multinomial regression model). Sex differences in immune responses to pathogens and vaccines are well documented [28,29], with females generally developing significantly higher levels of humoral immunity than males, for example, after vaccination against influenza [30] or hepatitis B [31]. Sex differences in immune response have been explained by the hormonal milieu that can have profound effects on B-cell activity regulation, antibody production and vaccine efficacy, with an elevated level of estradiol associated with higher vaccine-induced immunity [32]. Current data on SARS-CoV-2 vaccination are not sufficient to draw definitive conclusions on sex differences in vaccine efficacy, as reported in a recent review of 41 experimental and observational studies in which Vassallo, A. et al. explained the failure to recognize the gender difference through a lack of attention to sex/gender-specific evidence in the design of COVID-19 vaccine research [33].

Finally, although our study does not provide data on the effect of different ethnic groups, recent US [34] and UK [23] studies highlighted a greater seropositivity in non-white groups after BNT162b2 mRNA vaccination.

Among the limitations of the study, an important issue that warrants mention and requires some caution in the interpretation is that we did not perform sequential serum tests for viral neutralizing antibodies titer, nor IgA antibody levels, or the magnitude of T-cells elicited by the BNT162b2 mRNA vaccine. However, the quantitative antibodies assay targeting the spike RBD of SARS-CoV-2 that was used is recognized as a valid surrogate [35,36] that reflects vaccine response. However, since the study protocol did not include patients’ follow-ups during the weeks in which blood samples were taken, or after the full vaccination course, our results did not allow us to provide results on the actual protective capacity against SARS-CoV-2 infections linked to seroconversion. To minimize this concern, we also stratified seropositive subjects into two groups so as to be able to identify those at greatest risk of developing a non-neutralizing viral response. 

Lastly, we did not measure antibodies targeting the viral N protein (which is contained in the whole virus), at samples performed at T1 and at T2, so our study does not allow us to definitively conclude if the magnitude of the responses were due to the vaccination or to the eventual infection that occurred after the administration of the first or the second vaccination dose.

In summary, assessing the efficacy of a vaccine is always a complex process, but proved to be particularly so in the case of SARS-CoV-2, since the fundamental understanding of the pathogen is rapidly evolving. Serology testing has been used in the past to support immunization policies and strategies across a variety of vaccine-preventable diseases, providing data on several aspects, such as long-term persistence of protection and the need of a booster dose [31,37], or the vaccine’s efficacy in a specific population [38]. Moreover, serology testing has already shown its usefulness since the beginning of the pandemic, before the introduction of COVID-19 vaccines, in the surveillance of the cumulative spread of SARS-CoV-2 within populations by determining the actual number of infections through the serological diagnosis of asymptomatic infections [39]. 

Taking into account that to date, the protective antibody titers are not yet defined, and only the continuous surveillance of the population will allow us to clarify aspects of the vaccine’s effectiveness in terms of the duration of protection and the need for booster doses (albeit in light of the above recognized limitations), it is possible to hypothesize as to the usefulness of serological evaluation, in the subgroup of frail subjects after their first dose to implement personalized vaccination protocols. The intensification of the vaccination protocol, through the introduction of additional doses (third or even fourth), or with high dose vaccines, e.g., through the administration of two vaccine doses simultaneously, appears to be the more simple and feasible strategy for strengthening protection against SARS-CoV-2 in the event of a low response. Previous studies have demonstrated the advantageousness of an increased dose strategy for improving HBV vaccination response in hemodialysis patients. For subjects with normal immune statuses, HBV vaccination comprises three intramuscular doses, with 20 µg/mL of HBsAg protein, and for hemodialysis patients, a 0-1-2- and 6-month schedule with double doses of commercially available vaccine is recommended [40]. Similar results were obtained from studies conducted on different categories of immunocompromised patients vaccinated against influenza, as reported in a recent meta-analysis [41]. 

## 5. Conclusions

The results of this large prospective study demonstrated that the BNT162b2 mRNA vaccine is highly immunogenic and that the antibody level after two vaccine doses was significantly related to the antibody titer developed after the first vaccine dose. In addition, the stratification of the seroconversion level allowed us to identify not only subjects with a high risk of non-response, but also those who may not develop a viral neutralizing titer, and for whom alternate protection strategies could be developed. Some caution is required in the interpretation of antibody results and analyzing any subsequent behavior changes. Large-scale studies will be required to better understand the extent to which antibody titers are associated with vaccine protection.

## Figures and Tables

**Figure 1 vaccines-09-01494-f001:**
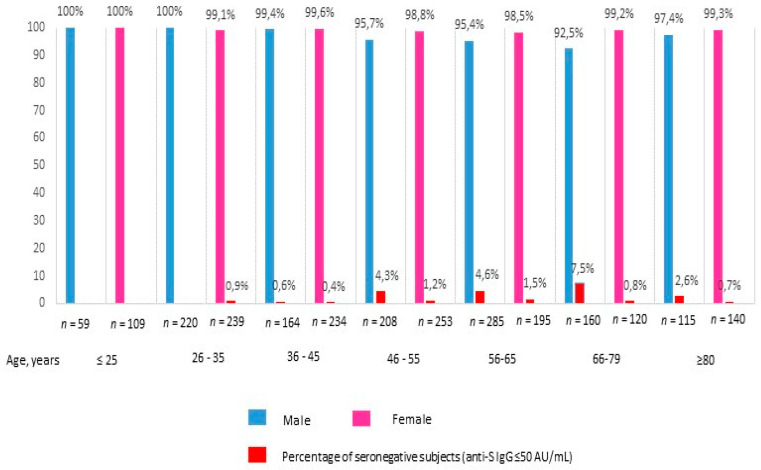
Percentage of seroconversion (anti-S IgG ≥ 50 AU/mL) 30 days after the conclusion of full course of BTN162b2 vaccination, by age and sex.

**Table 1 vaccines-09-01494-t001:** Characteristics of subjects tested 30 days after full course of vaccination stratified by anti-SARS-CoV-2 titer.

Characteristic	Total		Anti-S IgG (≤50 AU/mL)		Anti-S IgG (51–4159 AU/mL)		Anti-S IgG (≥4160 AU/mL)	
	N (2591)	%	N (50)	1.9%	N (602)	23.2%	N (1939)	74.8%
**Gender**								
Male Female	1208 1383	46.6 53.4	38 12	3.1 0.9	340 262	28.2 18.9	830 1109	68.7 80.2
			**χ^2^ = 52.1, 2 df, *p* < 0.001**					
**Age at testing, years**								
Mean ± SD	50.3 ± 18.5		61.2 ± 13.1		62.3 ± 17.9		46.3 ± 17.1	
			**F = 211.08, 2 df, *p <* 0.001**					
**Risk category**								
Healthy subjects Frail subjects Over 80 years	1910 427 25	73.7 16.5 9.8	2 44 4	0.1 10.3 1.6	288 178 136	15.1 41.7 53.5	1620 205 114	84.8 48 44.9
			**Fisher exact *p <* 0.001**					
**Anti-S IgG at T1(1641) ***								
Seronegative (≤50 AU/mL) Seropositive (>50 AU/mL)	162 1479	9.9 90.1	25 2	15.4 0.1	110 241	67.9 16.3	27 1236	16.7 83.6
			**Fisher exact *p <* 0.001**					

* Number of subjects who underwent blood samples for antibody testing at the time of their second vaccine dose (T1) is indicated in brackets. SD: standard deviation; df: degree of freedom.

**Table 2 vaccines-09-01494-t002:** Results of the multinomial regression analysis estimating predictors of the anti-SARS-CoV-2 titer measured after full course of vaccination.

Likelihood = −656.76; χ^2^ = 652.3 (10 df); *p* < 0.0001; No. of Observation = 1641				
Outcome: Anti-SARS-CoV-2 Titer Measured after Full Course of Vaccination				
	≤50 AU/mL		51–4159 AU/mL	
	**RRR (95% CI)**	***p* Value**	**RRR (95% CI)**	***p* Value**
**Gender** Female Male as reference	0.95 (0.32–2.81)	0.931	0.81 (0.61–1.07)	0.144
**Age at enrollment, continuous**	0.96 (0.92–1.00)	0.064	1.01 (0.99–1.02)	0.098
**Risk category**				
Frail subjects	4.21 (0.70–25.1)	0.115	1.07 (0.69–1.65)	0.744
Over 80	1.03 (0.08–13.7)	0.981	1.18 (0.64–2.16)	0.594
Healthy subjects as reference				
**Anti-S IgG at T1** Seropositive (>50 AU/mL) Seronegative (≤50 AU/mL) as reference	0.94 (0.90–0.97)	**<0.001**	0.99 (0.98–0.99)	**<0.001**

## Data Availability

The data presented in this study are available on request from the corresponding author.

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
