# Peer review of "Serological Response to SARS-CoV-2 Messenger RNA Vaccine: Real-World Evidence from Italian Adult Population"

_vaccines, 2021, doi:10.3390/vaccines9121494_

Round 1
Reviewer 1 Report
In the manuscript of Papadopoli and colleagues, the authors describe a prospective study - performed between December 2020 and May 2021 – that referred the entity of antibodies induced by the BNT162b2 vaccine against SARS-CoV-2 in subjects identified as priority groups for SARS- CoV-2 vaccination by the Italian Ministry of Health (healthcare workers, subjects aged 80 years and more and frail patients) and teaching and administrative University staff.
The results demonstrate that the BTN162b2 mRNA vaccine induces seroconversion in most of the large cohort of vaccinated subjects. Three blood samples were taken for each participant: at the time of the first vaccine dose (T0) (the presence of IgM and / or IgG titers equal to or greater than the cut-off classified as seropositive and excluded from the study); after 21 days, at the time of the second vaccine dose (T1); and 30 days after this last dose (T2).
Participants were stratified, according to their IgG anti-S titer at T2, into three groups: seronegative (≤50 AU/ml), low-intermediate responders (51-4,159 AU/ml) and high responders (≥4,160 AU/ml); The latter value, as reported Ebinger et coll, was used to evaluate the capacity of the recipients to achieve an IgG anti-S titer adequate to develop viral neutralizing antibodies.
The authors observed that the only seroconversion predictor the after the two administrated vaccination doses could be the IgG anti-S titer measured at T1. Seronegative (≤50 AU/ml) subjects at T1 were more likely to remain seronegative at T2 or to develop a low-intermediate (51-4,159 AU/ml) IgG anti-S titer.
The manuscript is very satisfying and exhaustive with respect to the topic and updates the knowledge on an important topic that has emerged in the last year, concerning the usefulness in orienting vaccine management strategies. The authors identify a high-risk group that may not develop a viral neutralizing response even in the presence of seroconversion, this is important for creating alternative prevention and protection strategies.
The authors have full and recognized experience in this field.
Minor revision
The relevance of having a high antibody titer to obtain a neutralizing response to viral infection has been extensively described by the authors. However, it would be interesting to add some information regarding the behaviour of subjects with high antibody titer (≥4.160 AU / ml) towards SARS-CoV-2 infection, to better understand the real efficacy of the high titer.
Author Response
The relevance of having a high antibody titer to obtain a neutralizing response to viral infection has been extensively described by the authors. However, it would be interesting to add some information regarding the behavior of subjects with high antibody titer (≥4.160 AU / ml) towards SARS-CoV-2 infection, to better understand the real efficacy of the high titer.
In response to the point, we agree with the reviewer comment that it would be interesting to add information regarding the behavior of subjects with high antibody titer (≥4.160 AU / ml) towards SARS-CoV-2 infection, but in the limit section of the discussion (page 7, lines 254-257 ) we added that since the study protocol did not include patients followup in the weeks among blood samples or after the full vaccination course, our results did not allow to provide results on the actual protective capacity against SARS-CoV2 infections linked to seroconversion.
Reviewer 2 Report
This study examined antibody levels after the first and second vaccination with BTN162b2 mRNA. The authors found the vaccine was highly immunogeneic and antibody levels generated after the second vaccine dose were significantly related to antibody titers after the first vaccine dose. Subject subgroups (ie the elderly) were also identified that had high frequencies of participants that did not develop a viral neutralising titer.
The methods, results and interpretation of the results are clearly outlined and straightforward. However, there are some areas of the manuscript that warrantee further clarification and the discussion could be expanded to highlight whether the findings are unique and/or comparable to other relevant studies in the literature. Also it would be useful to expand on the clinical implications of the findings.
Issues to address:
Results
Page 3, Line 117 and Table 1. It is confusing that eligible subjects “provided at least one blood sample for antibody testing”, whereas in Table 1 the characteristics of participants are described 30 days full vaccination (n=2591). As only a proportion of participants provided a blood sample at T2 (n= 1725, line 131, page 4), Table 1 is confusing as characteristics of 2591 participants after full vaccination are given but shouldn’t this be restricted to 1725 participants where data is available. Could the authors please explain.
Figure 1, recommend stratifying ages further considering a subset of participants aged ≥80 years were recruited. That is, instead of ≥66, this could be 66-75 and ≥75. Suggest adding this figure analyses data from n=1725 participants that provided a blood sample after the second vaccination.
Discussion
Paragraph 1 of discussion is very long. Consider dividing into >1 paragraphs. Perhaps talking about the implications of the results in the second paragraph (“Our results could be useful…)
Line 6, line 184. While the authors mention that Ab titers were in line with “previous studies carried out in the real-world”, as this is an Italian-based study it would be interesting how the Ab titer results of this study compare with finding from other Italian studies or studies from countries. That is, are the titers in line with what have previously been reported in the literature? Also does ethnicity impact on Ab titers (that is, are titers from other countries higher or lower)?
It would also be of interest to know whether this is the first study to identify whether antibody levels after the second vaccine dose were found to be significantly related to antibody titers after the first vaccine dose. Suggest including a comment related to this in the discussion.
Page 6, line 208. The authors are to be commended for addressing the limitations of the study. In particular, that they did not test for neutralizing Abs titers or T cell responses.
In the discussion it would be interesting to see whether the phenomenon of low seroconversion in males has been found in other studies analysing responses to the same vaccine or another vaccine and whether there is a biologically feasible or theoretical explanation for this observation. Could it be the result of a potential confounder such as comorbidities not being included in multinomial regression analyses. Further to this, perhaps it is worth explain what it means “suffering from chronic diseases” (page 4, line 148) as some comorbidities may not be considered a chronic disease but could affect a person’s immune response to a vaccine.
As part of their conclusion, the authors are advised to comment on the clinical implications of study findings. That is, do they recommend that certain high-risk subgroups (eg immunocompromised) have Ab titres measured after their first dose (or second?) to determine whether they have developed a viral neutralising titer or are findings merely for interest?
Page 6, line 234 – “alternate protection strategies could be developed”. There are several Covid-19 vaccines, would any be suitable? Would a T-cell mediated vaccine be better? If so, how would a different strategy boost immune responses, especially in the frail/immunosuppressed? Rather than mention this in the conclusion, suggest a separate paragraph with a few sentences could be devoted to discussing this important topic.
Minor comments
Page 6, line 225 – correct typo “jet”
There are a few grammar errors
Page 6, line 190 – consider deleting “only”
Page 6, line 194 - “As regard frail status” should perhaps read as “With regards to frail status”
Author Response
Page 3, Line 117 and Table 1. It is confusing that eligible subjects “provided at least one blood sample for antibody testing”, whereas in Table 1 the characteristics of participants are described 30 days full vaccination (n=2591). As only a proportion of participants provided a blood sample at T2 (n= 1725, line 131, page 4), Table 1 is confusing as characteristics of 2591 participants after full vaccination are given but shouldn’t this be restricted to 1725 participants where data is available. Could the authors please explain.
In response to the point, we have clarified in the results section (page 4, lines 137-139) number of subjects which provided blood samples scheduled by the study protocol 1,725 subjects (66.6%, derived by 1,725/2,591) provided sample at T0; 1,641 subjects provided sample at T1 (63.3% derived by 1,641/2,591), and 2,591 subjects provided sample at T2 (Total sample), 1,215 were the subjects which provided all three blood samples.
Figure 1, recommend stratifying ages further considering a subset of participants aged ≥80 years were recruited. That is, instead of ≥66, this could be 66-75 and ≥75. Suggest adding this figure analyses data from n=1725 participants that provided a blood sample after the second vaccination.
In response to the point, we have modified Figure 1 according to the suggestion.
Paragraph 1 of discussion is very long. Consider dividing into >1 paragraphs. Perhaps talking about the implications of the results in the second paragraph (“Our results could be useful…)
We modified the first paragraph of the discussion section according to the suggestion.
Line 6, line 184. While the authors mention that Ab titers were in line with “previous studies carried out in the real-world”, as this is an Italian-based study it would be interesting how the Ab titer results of this study compare with finding from other Italian studies or studies from countries. That is, are the titers in line with what have previously been reported in the literature? Also does ethnicity impact on Ab titers (that is, are titers from other countries higher or lower)?
In response to the point, we added in the results section (page 4 lines 143-144) that we found a percentage of seroconversion (anti-S IgG >50 AU/mL) of 99.8% and, in the discussion section (page 6 lines 200-206), we compared our findings with results of previous Italian studies and with those from other countries. Moreover, although our study does not provide data on the effect of different ethnic groups, in the discussion section (page 7 line 246-248) we added a comment concern this issue.
It would also be of interest to know whether this is the first study to identify whether antibody levels after the second vaccine dose were found to be significantly related to antibody titers after the first vaccine dose. Suggest including a comment related to this in the discussion.
As suggested we added a comment in the discussion section (page 5, lines 185-186).
Page 6, line 208. The authors are to be commended for addressing the limitations of the study. In particular, that they did not test for neutralizing Abs titers or T cell responses.
In response to the point, we have recognized this limitation and the need to use some caution in the interpretation of the results although, we have used a quantitative antibodies assay that is recognized as a valid surrogate of the vaccine response.
In the discussion it would be interesting to see whether the phenomenon of low seroconversion in males has been found in other studies analysing responses to the same vaccine or another vaccine and whether there is a biologically feasible or theoretical explanation for this observation. Could it be the result of a potential confounder such as comorbidities not being included in multinomial regression analyses. Further to this, perhaps it is worth explain what it means “suffering from chronic diseases” (page 4, line 148) as some comorbidities may not be considered a chronic disease but could affect a person’s immune response to a vaccine.
In response to the point, we commented, in the discussion section (page 6-7, lines 231-245) the role of sex in the antibody responses to vaccines. Moreover, in the results section (page 4, line 148) we have removed the sentence “suffering from chronic diseases”. Also, in the methods section, statistical analysis paragraph (page 3, lines 112-118) we have thoroughly defined characteristics of the subjects included in the subgroups of the risk categories and, in particular, what we mean as “frail subjects”.
As part of their conclusion, the authors are advised to comment on the clinical implications of study findings. That is, do they recommend that certain high-risk subgroups (eg immunocompromised) have Ab titres measured after their first dose (or second?) to determine whether they have developed a viral neutralising titer or are findings merely for interest?
In response to the point we added a comment on the role of the serological evaluation after first dose in frail subjects to implement personalized vaccination protocols (discussion section, page 7, lines 274-279).
Page 6, line 234 – “alternate protection strategies could be developed”. There are several Covid-19 vaccines, would any be suitable? Would a T-cell mediated vaccine be better? If so, how would a different strategy boost immune responses, especially in the frail/immunosuppressed? Rather than mention this in the conclusion, suggest a separate paragraph with a few sentences could be devoted to discussing this important topic.
As requested, a more detailed hypothesis about the intensification of anti SARS-Cov2 vaccination strategy was provided in the discussion section (page 7, lines 279-290).
Page 6, line 225 – correct typo “jet”
In response to the point, in the Discussion section, we replaced “jet” with “already”.
Page 6, line 190 – consider deleting “only”
As suggested, in the Discussion section, we deleted “only”
Page 6, line 194 - “As regard frail status” should perhaps read as “With regards to frail status”
As suggested, in the Discussion section, we replaced “As regard frail status” with “With regards to frail status”.
Page 3, Line 117 and Table 1. It is confusing that eligible subjects “provided at least one blood sample for antibody testing”, whereas in Table 1 the characteristics of participants are described 30 days full vaccination (n=2591). As only a proportion of participants provided a blood sample at T2 (n= 1725, line 131, page 4), Table 1 is confusing as characteristics of 2591 participants after full vaccination are given but shouldn’t this be restricted to 1725 participants where data is available. Could the authors please explain.
In response to the point, we have clarified in the results section (page 4, lines 137-139) number of subjects which provided blood samples scheduled by the study protocol 1,725 subjects (66.6%, derived by 1,725/2,591) provided sample at T0; 1,641 subjects provided sample at T1 (63.3% derived by 1,641/2,591), and 2,591 subjects provided sample at T2 (Total sample), 1,215 were the subjects which provided all three blood samples.
Figure 1, recommend stratifying ages further considering a subset of participants aged ≥80 years were recruited. That is, instead of ≥66, this could be 66-75 and ≥75. Suggest adding this figure analyses data from n=1725 participants that provided a blood sample after the second vaccination.
In response to the point, we have modified Figure 1 according to the suggestion.
Paragraph 1 of discussion is very long. Consider dividing into >1 paragraphs. Perhaps talking about the implications of the results in the second paragraph (“Our results could be useful…)
We modified the first paragraph of the discussion section according to the suggestion.
Line 6, line 184. While the authors mention that Ab titers were in line with “previous studies carried out in the real-world”, as this is an Italian-based study it would be interesting how the Ab titer results of this study compare with finding from other Italian studies or studies from countries. That is, are the titers in line with what have previously been reported in the literature? Also does ethnicity impact on Ab titers (that is, are titers from other countries higher or lower)?
In response to the point, we added in the results section (page 4 lines 143-144) that we found a percentage of seroconversion (anti-S IgG >50 AU/mL) of 99.8% and, in the discussion section (page 6 lines 200-206), we compared our findings with results of previous Italian studies and with those from other countries. Moreover, although our study does not provide data on the effect of different ethnic groups, in the discussion section (page 7 line 246-248) we added a comment concern this issue.
It would also be of interest to know whether this is the first study to identify whether antibody levels after the second vaccine dose were found to be significantly related to antibody titers after the first vaccine dose. Suggest including a comment related to this in the discussion.
As suggested we added a comment in the discussion section (page 5, lines 185-186).
Page 6, line 208. The authors are to be commended for addressing the limitations of the study. In particular, that they did not test for neutralizing Abs titers or T cell responses.
In response to the point, we have recognized this limitation and the need to use some caution in the interpretation of the results although, we have used a quantitative antibodies assay that is recognized as a valid surrogate of the vaccine response.
In the discussion it would be interesting to see whether the phenomenon of low seroconversion in males has been found in other studies analysing responses to the same vaccine or another vaccine and whether there is a biologically feasible or theoretical explanation for this observation. Could it be the result of a potential confounder such as comorbidities not being included in multinomial regression analyses. Further to this, perhaps it is worth explain what it means “suffering from chronic diseases” (page 4, line 148) as some comorbidities may not be considered a chronic disease but could affect a person’s immune response to a vaccine.
In response to the point, we commented, in the discussion section (page 6-7, lines 231-245) the role of sex in the antibody responses to vaccines. Moreover, in the results section (page 4, line 148) we have removed the sentence “suffering from chronic diseases”. Also, in the methods section, statistical analysis paragraph (page 3, lines 112-118) we have thoroughly defined characteristics of the subjects included in the subgroups of the risk categories and, in particular, what we mean as “frail subjects”.
As part of their conclusion, the authors are advised to comment on the clinical implications of study findings. That is, do they recommend that certain high-risk subgroups (eg immunocompromised) have Ab titres measured after their first dose (or second?) to determine whether they have developed a viral neutralising titer or are findings merely for interest?
In response to the point we added a comment on the role of the serological evaluation after first dose in frail subjects to implement personalized vaccination protocols (discussion section, page 7, lines 274-279).
Page 6, line 234 – “alternate protection strategies could be developed”. There are several Covid-19 vaccines, would any be suitable? Would a T-cell mediated vaccine be better? If so, how would a different strategy boost immune responses, especially in the frail/immunosuppressed? Rather than mention this in the conclusion, suggest a separate paragraph with a few sentences could be devoted to discussing this important topic.
As requested, a more detailed hypothesis about the intensification of anti SARS-Cov2 vaccination strategy was provided in the discussion section (page 7, lines 279-290).
Page 6, line 225 – correct typo “jet”
In response to the point, in the Discussion section, we replaced “jet” with “already”.
Page 6, line 190 – consider deleting “only”
As suggested, in the Discussion section, we deleted “only”
Page 6, line 194 - “As regard frail status” should perhaps read as “With regards to frail status”
As suggested, in the Discussion section, we replaced “As regard frail status” with “With regards to frail status”.
Page 3, Line 117 and Table 1. It is confusing that eligible subjects “provided at least one blood sample for antibody testing”, whereas in Table 1 the characteristics of participants are described 30 days full vaccination (n=2591). As only a proportion of participants provided a blood sample at T2 (n= 1725, line 131, page 4), Table 1 is confusing as characteristics of 2591 participants after full vaccination are given but shouldn’t this be restricted to 1725 participants where data is available. Could the authors please explain.
In response to the point, we have clarified in the results section (page 4, lines 137-139) number of subjects which provided blood samples scheduled by the study protocol 1,725 subjects (66.6%, derived by 1,725/2,591) provided sample at T0; 1,641 subjects provided sample at T1 (63.3% derived by 1,641/2,591), and 2,591 subjects provided sample at T2 (Total sample), 1,215 were the subjects which provided all three blood samples.
Figure 1, recommend stratifying ages further considering a subset of participants aged ≥80 years were recruited. That is, instead of ≥66, this could be 66-75 and ≥75. Suggest adding this figure analyses data from n=1725 participants that provided a blood sample after the second vaccination.
In response to the point, we have modified Figure 1 according to the suggestion.
Paragraph 1 of discussion is very long. Consider dividing into >1 paragraphs. Perhaps talking about the implications of the results in the second paragraph (“Our results could be useful…)
We modified the first paragraph of the discussion section according to the suggestion.
Line 6, line 184. While the authors mention that Ab titers were in line with “previous studies carried out in the real-world”, as this is an Italian-based study it would be interesting how the Ab titer results of this study compare with finding from other Italian studies or studies from countries. That is, are the titers in line with what have previously been reported in the literature? Also does ethnicity impact on Ab titers (that is, are titers from other countries higher or lower)?
In response to the point, we added in the results section (page 4 lines 143-144) that we found a percentage of seroconversion (anti-S IgG >50 AU/mL) of 99.8% and, in the discussion section (page 6 lines 200-206), we compared our findings with results of previous Italian studies and with those from other countries. Moreover, although our study does not provide data on the effect of different ethnic groups, in the discussion section (page 7 line 246-248) we added a comment concern this issue.
It would also be of interest to know whether this is the first study to identify whether antibody levels after the second vaccine dose were found to be significantly related to antibody titers after the first vaccine dose. Suggest including a comment related to this in the discussion.
As suggested we added a comment in the discussion section (page 5, lines 185-186).
Page 6, line 208. The authors are to be commended for addressing the limitations of the study. In particular, that they did not test for neutralizing Abs titers or T cell responses.
In response to the point, we have recognized this limitation and the need to use some caution in the interpretation of the results although, we have used a quantitative antibodies assay that is recognized as a valid surrogate of the vaccine response.
In the discussion it would be interesting to see whether the phenomenon of low seroconversion in males has been found in other studies analysing responses to the same vaccine or another vaccine and whether there is a biologically feasible or theoretical explanation for this observation. Could it be the result of a potential confounder such as comorbidities not being included in multinomial regression analyses. Further to this, perhaps it is worth explain what it means “suffering from chronic diseases” (page 4, line 148) as some comorbidities may not be considered a chronic disease but could affect a person’s immune response to a vaccine.
In response to the point, we commented, in the discussion section (page 6-7, lines 231-245) the role of sex in the antibody responses to vaccines. Moreover, in the results section (page 4, line 148) we have removed the sentence “suffering from chronic diseases”. Also, in the methods section, statistical analysis paragraph (page 3, lines 112-118) we have thoroughly defined characteristics of the subjects included in the subgroups of the risk categories and, in particular, what we mean as “frail subjects”.
As part of their conclusion, the authors are advised to comment on the clinical implications of study findings. That is, do they recommend that certain high-risk subgroups (eg immunocompromised) have Ab titres measured after their first dose (or second?) to determine whether they have developed a viral neutralising titer or are findings merely for interest?
In response to the point we added a comment on the role of the serological evaluation after first dose in frail subjects to implement personalized vaccination protocols (discussion section, page 7, lines 274-279).
Page 6, line 234 – “alternate protection strategies could be developed”. There are several Covid-19 vaccines, would any be suitable? Would a T-cell mediated vaccine be better? If so, how would a different strategy boost immune responses, especially in the frail/immunosuppressed? Rather than mention this in the conclusion, suggest a separate paragraph with a few sentences could be devoted to discussing this important topic.
As requested, a more detailed hypothesis about the intensification of anti SARS-Cov2 vaccination strategy was provided in the discussion section (page 7, lines 279-290).
Page 6, line 225 – correct typo “jet”
In response to the point, in the Discussion section, we replaced “jet” with “already”.
Page 6, line 190 – consider deleting “only”
As suggested, in the Discussion section, we deleted “only”
Page 6, line 194 - “As regard frail status” should perhaps read as “With regards to frail status”
As suggested, in the Discussion section, we replaced “As regard frail status” with “With regards to frail status”.